# LEGO: A Multi-agent Collaborative Framework with Role-playing and Iterative Feedback for Causality Explanation Generation

**Zhitao He**[1,2], **Pengfei Cao**[1,2], **Yubo Chen**[1,2], **Kang Liu**[1,2]
**Ruopeng li**[3], **Mengshu Sun**[3], **Jun Zhao**[1,2]

[1] The Laboratory of Cognition and Decision Intelligence for Complex Systems,
Institute of Automation, Chinese Academy of Sciences, Beijing, China
[2] School of Artificial Intelligence, University of Chinese Academy of Sciences, Beijing, China
[3] Ant Group, Hangzhou, China
{zhitao.he, pengfei.cao, yubo.chen, kliu, jzhao}@nlpr.ia.ac.cn

## Abstract

Causality Explanation Generation refers to generate an explanation in natural language given an initial cause-effect pair. It demands rigorous explicit rationales to demonstrate the acquisition of implicit commonsense knowledge, which is unlikely to be easily memorized, making it challenging for large language models since they are often suffering from spurious causal associations when they encounter the content that does not exist in their memory. In this work, we introduce **LEGO**, a Multi-agent Collaborative Framework with Role-playing and Iterative Feedback for causality explanation generation. Specifically, we exploit LLM as character malleable LEGO block and utilize role-playing to assign specific roles to five LLMs. We firstly devise a Fine-grained World Knowledge Integration Module to augment information about tasks for alleviating the phenomenon of spurious causal associations. Then, we leverage an Iterative Feedback and Refinement Module to improve the generated explanation by multi-aspect feedback. Extensive experiments on widely used WIKIWHY and e-CARE datasets show the superiority of our multi-agent framework in terms of reasoning about the causality among cause and effect.

## 1 Introduction

Causality explanation generation is a generative task that aims to explain why the given cause-effect pair is true using natural language. For example, given the cause clause $C$ and effect clause $E$ in Figure 1, a corresponding explanation $X$ needs to be generated is "*A monsoon was striking Ceylon or southern India at the time and the fleet of the seventh voyage of the Ming Treasure Voyage did not want to be caught in the storm*". Causality explanation generation can facilitate various applications, including explainable question answering (Yang et al., 2018), complex reasoning (Dalvi et al., 2021) and future event prediction (Zhou et al., 2022).

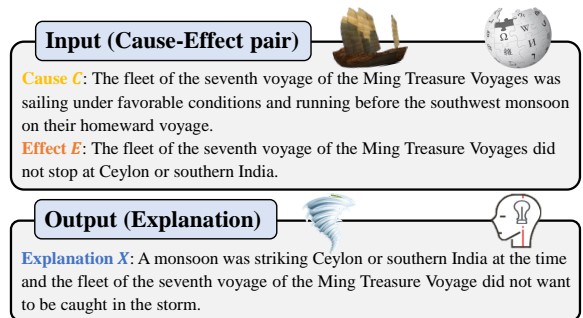

Figure 1: An example of causality explanation generation from the WIKIWHY dataset (Ho et al., 2023).

Despite the extensive applications of causality explanation generation, this task is highly challenging as it demands rigorous and explicit rationales to demonstrate that a model not only "knows" the causality in task but also "understands" the underlying mechanics of why that is the case. Previous studies (Du et al., 2022; Ho et al., 2023) leverage a single language model with in-context learning to tackle the task, however, language models often generate **spurious causal association** based on the given cause-effect pair, leading to deviation from the correct reasoning path. As depicted in Figure 2(a), when observing the information that "The fleet was sailing under favorable conditions" in cause $C$, GPT-3 and GPT-3.5 both generate the spurious causal association without exception, i.e., "*under favorable condition*" $\Rightarrow$ "*they were making good time*", which leads to incorrect reasoning directions and hinders language models from taking a step further to explore the causal mechanism.

Through in-depth analysis, we attribute this kind of errors to the limitation of **unidirectional** reasoning of large language models (LLMs). As illustrated in Figure 2(a), both GPT-3 and GPT-3.5 only conduct unidirectional reasoning based on the content of the cause. When encountering *favorable condition*, they directly associate it with *making good time*, which suggests that this spurious causal

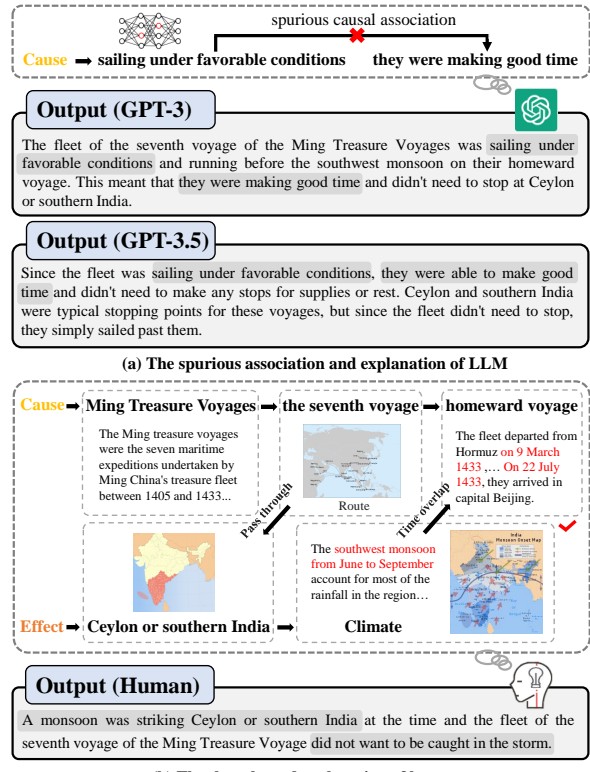

**(a) The spurious association and explanation of LLM**

**(b) The thought and explanation of human**

Figure 2: The explanation generated by large language models and human for the task mentioned above.

association appears to be ingrained in the parameters of large language models. As a result, they inevitably deviate from the correct reasoning path, overlook crucial information present in the effect, and fail to autonomously search for broader world knowledge beyond the given text.

Actually, it is widely accepted that human memory is characterized by its bidirectional associative and parallel processing capabilities (Kosko, 1988; Hattori and Hagiwara, 1995; Anderson and Bower, 2014). The ability of **bidirectional** reasoning may be helpful in alleviating the phenomenon of spurious causal associations. We demonstrate the bidirectional thought process required to arrive at the gold explanation in Figure 2(b). On the one hand, we begin our reasoning from the information on cause side and progressively search for "*Ming Treasure Voyages*", "*the seventh voyage*", and "*homeward voyage*". Eventually, we obtain the key fine-grained world knowledge: "The fleet departed from Hormuz on 9 March 1433, ... On 22 July 1433, they arrived in the capital Beijing". On the other hand, we reason from the information on effect side and search for "*Ceylon or southern India*", then associate it with the route of "the seventh

voyage" mentioned in cause side (pass through). Subsequently, from the regional climate data, we learn another key fine-grained world knowledge: "The southwest monsoon from June to September accounts for most of the rainfall in the region". Finally, by linking these two pieces of **fine-grained world knowledge** obtained by the bidirectional reasoning process, we discover that the return time of the seventh voyage coincides with the arrival of the southwest monsoon in southern India. Therefore, we can conclude ⇒ "*A monsoon was striking Ceylon or southern India at the time*". Combining this with **task-specific commonsense knowledge**, such as the fleet needed to avoid storms at sea, we can deduce ⇒ "*the fleet did not want to be caught in the storm*". In summary, to sucessful complete this task, the system needs the abilities of bidirectional reasoning and knowledge retrieval to effectively integrate fine-grained world knowledge. Furthermore, it necessitates the capacity for commonsense induction to augment task-specific commonsense knowledge.

Based on the detailed analysis above, we can be observed that it demands multiple abilities, such as bidirectional reasoning, knowledge retrieval, and commonsense induction for this complex task. Although LLMs demonstrate a wide range of capabilities, a single language model is unable to simultaneously provide all of them. Therefore, we propose a novel Multi-agent Collaborative Framework with Role-playing and Iterative Feedback (**LEGO**) to effectively combine the different abilities of multiple large language models for causality explanation generation. Specifically, we exploit LLMs as character malleable LEGO blocks and utilize role-playing to assign specific roles to the five LLMs. We firstly devise a *Fine-grained World Knowledge Integration Module* to augment information about tasks to alleviate the phenomenon of spurious causal associations. The module consists of three LLMs, two LLMs are designated as **Cause Analyst** and **Effect Analyst**, reasoning around Cause and Effect respectively to simulate the process of bidirectional inference, and posing questions to another LLM which acts as **Knowledge Master** to autonomously mine fine-grained world knowledge. Then, we leverage an *Iterative Feedback and Refinement Module* to improve explanation by multi-aspect feedback. The module utilizes one LLM as **Explainer** to generate an initial explanation, and iteratively receives Observation

feedback and Commonsense feedback from **Critic** LLM to refine its explanation.

Overall, the main contributions of this work can be summarized as follows: 1) We propose a novel multi-agent collaborative framework with role-playing and iterative feedback (LEGO) to effectively combine different abilities of multiple LLMs for causality explanation generation; 2) We devise a Fine-grained World Knowledge Integration Module to augment information about tasks through the interaction of three agent, i.e. Cause Analyst, Effect Analyst and Knowledge Master. We leverage an Iterative Feedback and Refinement Module to improve the generated explanation by multi-aspect feedback involving two LLMs, i.e. Explainer and Critic; 3) Extensive experiments on WIKIWHY and e-CARE show the superiority of our multi-agent framework in terms of reasoning about the causality among cause and effect.

## 2 Methodology

In this section, we introduce our multi-agent collaborative framework LEGO. As shown in Figure 2, our framework consists of two major components involving five LLMs: (1) Fine-grained World Knowledge Integration Module, which augments information about tasks through the interaction of three agents; (2) Iterative Feedback and Refinement Module, which utilizes one LLM serve as Explainer to generate an initial output, and iteratively receives Observation and Commonsense feedback from Critic LLM to refine its explanation.

### 2.1 Fine-grained World Knowledge Integration

We devise this module to precisely augment information about task for alleviating the phenomenon of spurious causal associations.

**Cause-Effect Analyst Role Assignment** After receiving the task, the Cause Analyst role and the Effect Analyst role will be assigned to two LLMs respectively by inception prompt (Li et al., 2023). In practice, a system message prompts are passed to the LLMs before the conversations start to assign LLMs with corresponding roles. We refer to the Cause Analyst message prompt is $P_C$ and that of the Effect Analyst is $P_E$. Let $L_1$ and $L_2$ denote two large language models, when the system message is passed to those large language models respectively, we obtain $\mathcal{M} \leftarrow L_1^{P_C}$ and $\mathcal{A} \leftarrow L_2^{P_E}$ which are referred to as the Cause Analyst role and the Effect

Analyst. In addition, The third agent, serves as the Knowledge Master for answering the questions of the reasoners, does not require a specific prompt.
**Reasoning Towards Causality** After the roles are assigned, the Cause Analyst $\mathcal{M}$ and Effect Analyst $\mathcal{A}$ will collaborate in reasoning about the fine-grained world knowledge by thought and action (Yao et al., 2022). The Cause Analyst is responsible for reasoning about the information in Cause and directing the conversation around the causality in task. Meanwhile, the Effect Analyst is designed to reason about the information in Effect and follow the reasoning trace of the Cause Analyst. One example of Effect Analyst reasoning about the task presented in Figure 1 is shown below:

```
Thought: I need to analyze from the
Effect. So I need to ask about Ceylon,
southern India and southwest monsoon.
Ask: [Ceylon, southern India and
southwest monsoon]
```

The Knowledge Master returns its Observation:

```
Observation: Ceylon, is an island country
located in the Indian Ocean, off the
southern coast of India. It experiences a
tropical climate and is greatly
influenced by the southwest monsoon...The
southwest monsoon typically occurs
between June and September each year...
```

Formally, we denote the Cause Analyst message obtained at time $t$ by $M_t$ and the corresponding observation from Knowledge Master is $O_{m_t}$, the Effect Analyst message is $A_t$ and the corresponding observation is $O_{a_t}$, the historical messages:

$$H_t = \{(M_i, O_{m_i}, A_i, O_{a_i})\}|_{i=0}^t \qquad (1)$$

At the next time step $t+1$, the Cause Analyst takes the historical conversation message set and creates a new message $M_{t+1}$, as shown:

$$M_{t+1} = \mathcal{M}(H_t) \qquad (2)$$

The assistant reasoning will respond with $A_{t+1}$:

$$A_{t+1} = \mathcal{A}(H_t, (M_{t+1}, O_{m_{t+1}})) \qquad (3)$$

We focus on the Observations obtained by two reasoners during their interaction, which encompass fine-grained knowledge about the task. These

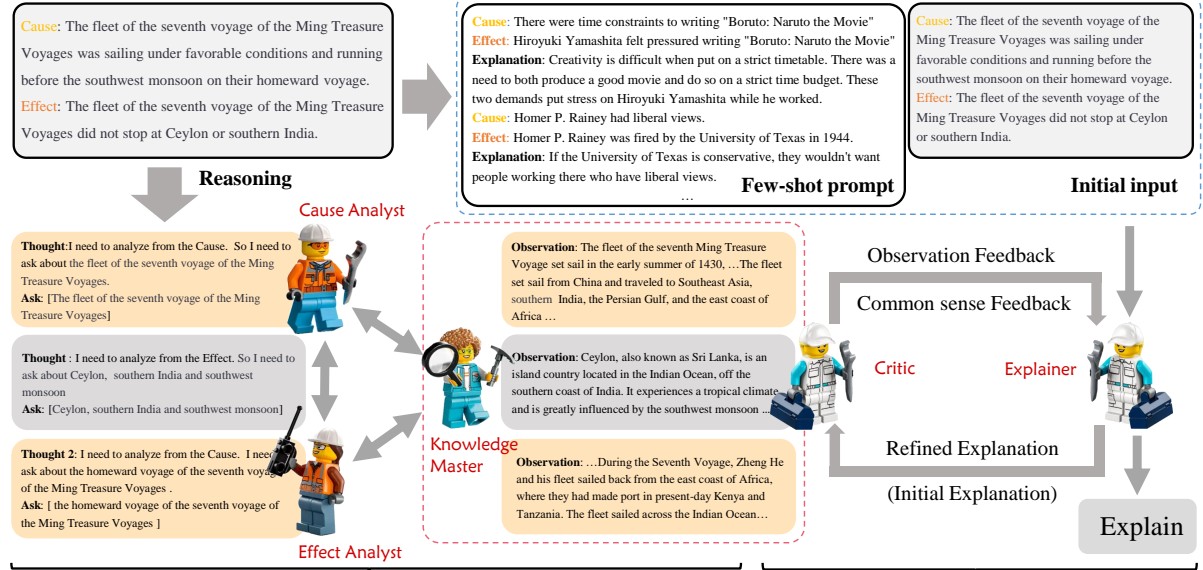

Figure 3: Overview of our multi-agent framework. It first augments information about current task through the interaction of three agents (left), then utilizes one LLM serve as Explainer to generate an initial output and iteratively refine its explanation by following the Observation and Commonsense feedback from Critic LLM (right).

Observations can aid the Explainer in alleviating the spurious causal associations. Moreover, the reasons why we use LLM as the Knowledge Master instead of searching on Knowledge base like Wikipedia are that 1) Wikipedia cannot accept free-form text queries from Analyst and the error analysis of ReAct (Yao et al., 2022) indicates 23% of the errors came from the search information returning empty or useless information; 2) Recently Yu et al. (2022) has demonstrated that the generated contextual documents more frequently contain the correct answer compared to the retrieved documents. Therefore, utilizing LLM to query a large amount of fine-grained knowledge is more efficient than traditional search methods. We will further analyze this in the experiment section.

**Inception Prompting**   Following Li et al. (2023), we utilize the inception prompts to declare roles to each LLM before the conversation begins. After the inception prompt is delivered to the LLM as a system message, the agent automatically assumes the corresponding role and interacts in the conversation with the way of thinking first and then action. Our inception prompt consists of Cause Analyst prompt $P_C$ and Effect Analyst $P_E$, which encompass role definitions, action spaces, and guidelines. We present part of the Cause Analyst prompt $P_C$ in Figure 4. The details of inception prompts are in Appendix E.

---

Never forget you are a Cause Analyst and I am a Effect Analyst. Never flip roles! … You need to reason ONLY in the following two ways:

1. Thought with necessary Ask:
   Thought: <YOUR_THOUGHT>
   Ask: <YOUR_QUESTION>

2. Thought without any Ask:
   Thought: <YOUR_THOUGHT>
   Ask: None

Here are some examples: …
You must give me ONLY one Thought at a time!...
Here is the task: <TASK>. Never forget our task!

Figure 4: Part of the Cause Analyst inception prompt.

## 2.2 Iterative Feedback and Refinement

In this section, one LLM serve as Explainer to generate output, and iteratively receives multi-aspect feedback from Critic LLM to refine its explanation.
**Multi-aspect Feedback**   Although LLMs can generate coherent outputs with in-context learning, they often fall short in addressing more intricate requirements. According to our error analysis[1] of GPT-3.5 on WIKIWHY dataset (Ho et al., 2023), the most prominent errors are the lack of common-sense knowledge and repetitive causal relationships (accounting for a combined 54%) and

---

[1]We random sample of 50 entries.

**Current Explanation ($y_i$):** The fleet was sailing under favorable conditions and running before the southwest monsoon on their homeward voyage. This means that they were trying to make good time to get back home and did not want to make any unnecessary stops that would slow them down. ✗ **Explainer**

**Observation Feedback**: The Explanation is not a simple concatenation of Cause and Effect, but ignores that the southwest monsoon typically arrives in Ceylon around May or June and lasts until September. **Critic**
**Commonsense Feedback**: Fleets need to pay close attention to the weather forecast and marine meteorological information, try to avoid being involved in the storm.

**Refined explanation ($y_{i+1}$):** The monsoon season typically arrives in Ceylon around May or June and lasts until September, which could have created risky sailing conditions for the fleet. Therefore, the fleet did not stop at these places to avoid being involved in the storm and continued their journey to get back . ✓

Figure 5: The explanation refinement process.

there are many research studies (Bai et al., 2022b; Yang et al., 2022) have demonstrated the success of multi-aspect feedback. Therefore, we decide to break critic feedback into observation and commonsense. Specifically, we utilize a LLM to act as a Critic to provide multi-aspect feedback on the explanation. Critic receives the explanation $y_i$ and Observations from previous stage, then provides Observation and Commonsense feedback to improve the explanation. The Observation feedback covers two aspects: 1) report on whether the explanation is repeating the cause-effect relation; 2) supplementary information based on Observation. The Commonsense feedback aims to present the commonsense knowledge required to explain the causality of the task.

**Iterative Refinement** The Explainer improves it output based on received feedback and previous generated output. The *Critc provides feedback $\Rightarrow$ Explainer refines explanation $\Rightarrow$ Critc provides feedback* loop can be applied multiple times. We set the number of iterations to a fixed number due to budget. One key aspect of Critic is the retention of a history of past experiences. This is achieved by appending the previous outputs to the prompt continuously. This allows Explainer to learn from past mistakes and avoid repeating them.

Figure 5 depicts the explanation refinement process. The current explanation exhibits the spurious causal association that "*under favorable condition*" $\Rightarrow$ "*they were trying to make good time*". The Critic, based on the Observations obtained by knowledge integration module, provides Obser-

vation feedback that the current explanation does not repeat cause-effect relation but overlooks the key information that "the southwest monsoon typically arrives in Ceylon around May or June and lasts until September." Furthermore, it raises the Commonsense feedback that "Fleets need to pay close attention to the weather forecast and marine meteorological information ..." After receiving the feedback, the Explainer becomes aware that the reason why the fleet did not stop was the monsoon striking the southern India at that time, and the fleet needed to avoid getting caught in the storm. It can be observed that the Explainer corrects the error of spurious causal association and generates a valid explanation by incorporating the mentioned commonsense.

## 3 Experiments

### 3.1 Datasets

We conduct extensive experiments on two datasets without training. **WIKIWHY** (Ho et al., 2023) a large-scale QA dataset built around explaining why an answer is true in natural language, which contains over 9,000 "why" question-answer-rationale triples, grounded on Wikipedia facts across a diverse set of topics. **e-CARE** (Du et al., 2022) is a large human-annotated explainable causal reasoning dataset, which contains over 21K causal reasoning questions, together with natural language formed explanations of the causal questions. Since the test set of e-CARE is not public and our method needs to call the OpenAI API which is costly. Therefore, we conduct experiments on the published validation set of e-CARE. We present data examples and other details in Appendix A.

### 3.2 Baselines

To assess the adaptability of our method, we use `text-davinci-002`, `text-davinci-003` and `gpt-3.5turbo` as the underlying language model, we also refer to these model as GPT-3 (Brown et al., 2020), InstructGPT (Ouyang et al., 2022), GPT-3.5. Furthermore, we conduct experiments with a wide range of LLMs and the ReAct (Yao et al., 2022) with Wikipedia search engine. For a fair comparison, we employ the same examples as Ho et al. (2023) to generate the initial explanation.

### 3.3 Evaluation Metrics

**Automatic Evaluation** We provide numerous examples from both datasets in Table 6 in the Ap-

| Experiments | Unordered | | | Ordered | | |
|---|---|---|---|---|---|---|
| | Precision | Recall | F1 | Precision | Recall | F1 |
| GPT-2 | 0.249 | 0.196 | 0.220 | 0.239 | 0.179 | 0.204 |
| GPT-3 | 0.347 | 0.388 | 0.366 | 0.307 | 0.355 | 0.329 |
| InstructGPT | 0.537 | 0.571 | 0.553 | **0.468** | 0.503 | 0.435 |
| GPT-3.5 | **0.609** | 0.620 | 0.615 | 0.467 | 0.557 | 0.508 |
| ReAct(GPT-3.5) | 0.597 | **0.669** | **0.631** | 0.455 | **0.595** | **0.516** |
| LEGO (InstructGPT) | | | | | | |
| *Iteration*1 | 0.565 | 0.586 | 0.575 | 0.486 | 0.521 | 0.503 |
| *Iteration*2 | **0.579** | **0.606** | **0.592** | **0.515** | **0.526** | **0.520** |
| *Iteration*3 | 0.556 | 0.583 | 0.569 | 0.479 | 0.515 | 0.496 |
| LEGO (GPT-3.5) | | | | | | |
| *Iteration*1 | 0.611 | 0.708 | 0.656 | **0.476** | 0.618 | **0.538** |
| *Iteration*2 | **0.624** | **0.714** | **0.666** | 0.464 | **0.634** | 0.536 |
| *Iteration*3 | 0.598 | 0.705 | 0.647 | 0.427 | 0.627 | 0.508 |

Table 1: Performance on WIKIWHY dataset. We conduct experiments on InstructGPT (`text-davinci-003`) and GPT-3.5 (`gpt-3.5-turbo`) as baselines and underlying language models respectively.

| Setting | Fine Grained Human Evaluation | | | | | | |
|---|---|---|---|---|---|---|---|
| | Correctness | Concision | Fluency | Validity | Win (↑) | Tie | Lose (↓) |
| GPT-2 | 0.100 | 0.880 | 0.860 | 0.520 | 0.040 | 0.040 | 0.920 |
| GPT-3 | 0.660 | 0.680 | 1.00 | 0.960 | 0.080 | 0.360 | 0.580 |
| InstructGPT | 0.673 | 0.700 | 1.00 | 0.973 | 0.166 | 0.353 | 0.480 |
| GPT-3.5 | 0.713 | 0.526 | 1.00 | 0.966 | 0.193 | 0.326 | 0.481 |
| ReAct (GPT-3.5) | 0.726 | 0.533 | 1.00 | 0.946 | 0.206 | 0.333 | 0.460 |
| LEGO (InstructGPT) | 0.706 | 0.660 | 0.986 | 0.973 | 0.186 | 0.340 | 0.474 |
| LEGO (GPT-3.5) | **0.753** | 0.506 | 1.00 | 0.953 | 0.226 | 0.313 | 0.448 |

Table 2: Human evaluation. We show the results after using the our framework. Overall correctness is marked on a binary scale to indicate an explanation is complete and satisfying or not.

| Model | AVG-BLEU | ROUGE-l | CEQ | Human |
|---|---|---|---|---|
| InstructGPT | 19.72 | 30.49 | 0.023 | 0.464 |
| GPT-3.5 | 22.75 | 32.82 | 0.027 | 0.526 |
| ReAct (GPT-3.5) | 23.26 | 33.05 | 0.028 | 0.513 |
| LEGO(InstructGPT) | 21.32 | 32.54 | 0.028 | 0.536 |
| LEGO(GPT-3.5) | **25.23** | **36.18** | **0.032** | **0.560** |

Table 3: Performance on e-CARE dataset. We used a binary scale (correct/incorrect) for human evaluation and report the proportion of correct evaluations. For comparison, the human-generated CEQ score is 0.038.

pendix 5. It is worth noting that the explanations in e-CARE focus on elaborating the causal facts at a conceptual level, typically encompassing only one conceptual sentence. For instance, "Emoticons are combinations of characters used to represent various emotions." (example 4 of e-CARE). Therefore, to maintain consistency with previous works, we follow (Du et al., 2022) to leverage average-BLEU (n=4) (Papineni et al., 2002), ROUGE-l (Lin, 2004)

and CEQ (Causal Explanation Quality) (Du et al., 2022) as metrics.

In contrast, explanations in WikiWhy exhibit two structures: multi-hop step sequences and rationale sets, rendering them more instantiated. According to statistics, the average length of explanations in this dataset is 1.5 sentences (Table 4), with some extending up to 6 sentences. Additionally, there is a fixed order among the sentences. Consequently, the paper of WikiWhy introduces both ordered and unordered evaluation to compare the ideas contained in the predictions and references. Specifically, we follow Ho et al. (2023) to use unordered evaluation and ordered evaluation. The former aims to compare the ideas contained in the predictions and references, and the latter tends to penalize incorrectly ordered explanations for the structure of multi-hop explanations. Details of the unordered and ordered evaluation can be found in Appendix B.

**Human Evaluation**    To ensure consistency with

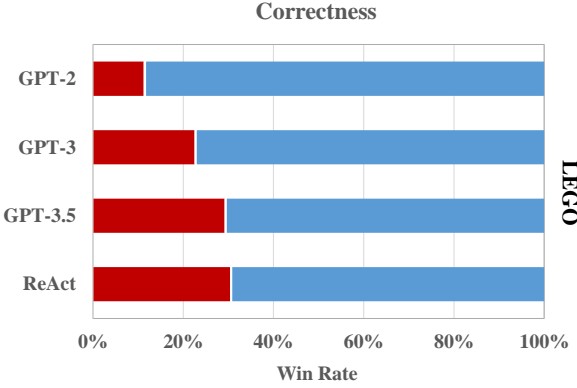

Figure 6: LEGO (GPT-3.5) vs baselines on the human evaluation metric Correctness.

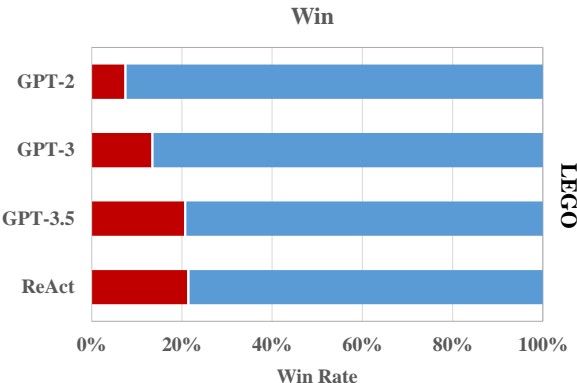

Figure 7: LEGO (GPT-3.5) vs baselines on the human evaluation metric Win

prior works (Ho et al., 2023), we present a panel of three graduate students a random sample of 50 entries from each setting and the following binary True/False criteria guidelines: 1) *Correctness*: Is the explanation both complete and satisfying? 2) *Concision*: Does the explanation say everything it needs to say and nothing more? 3) *Fluency*: Is the explanation writing fluent? 4) *Validity*: Does the explanation make logical sense? 5) *Win/Tie/Lose*: Compare the generated explanation against the provided reference. The three annotators work independently on randomly selected samples. The human evaluations were conducted via spreadsheets. We randomly shuffled the golden reference and the generated candidates, we don't reveal which column is the golden reference and which is the generated candidate in the evaluation setup. Details of human evaluation can be found in Appendix B.

### 3.4 Main Results

Our main results of WIKIWHY dataset are provided in Table 1. We can observe that our framework significantly improves the quality of explanations. For InstructGPT based setting, there is a 3.9% improvement in unordered f1 score by using our framework; similarly, for GPT-3.5 based setting, there is a 5.1% improvement. This establishes that our method is effective for different language models. The corresponding human evaluation experiments presented in Table 2. 75.3% of the explanations generated by our framework with GPT-3.5 are judged to be satisfactory, significantly higher than other baselines. It should be noted that the results in Table 2 just intend to count the number of samples in the current generated explanations that meet these metrics (e.g. "Correctness"), which cannot intuitively reflect the advantages of our model.

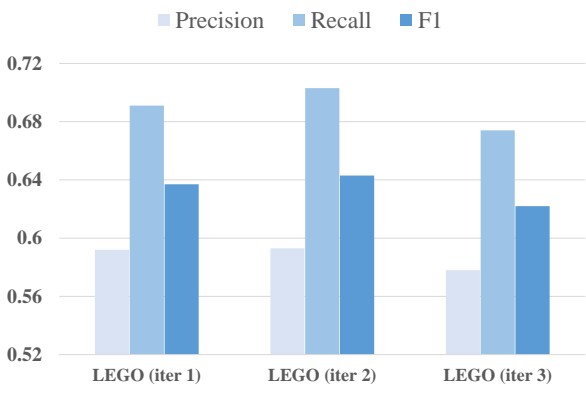

Figure 8: Performance change of our framework without knowledge integration module.

Thus by using the explanations of LEGO (GPT-3.5) as references, we compared different baselines against LEGO (GPT-3.5) separately. This was done to visually demonstrate the superiority LEGO (GPT-3.5), as shown in Figure 6 and 7. The quality of the explanations we generate is significantly better than that generated by baselines under different metrics. More detail can be found in Table 8 in Appendix.

Furthermore, We show the complete cases in Appendix D. We demonstrate the experimental results of e-CARE in Table 3. The produced explanations by our framework with GPT-3.5 judged to be satisfactory is 56% and achieve a CEQ score close to that of human-generated explanations. We consider this is due to the e-CARE dataset focusing on elaborating the causal facts at a conceptual level and there is no need to integrate complex fine-grained world knowledge. We further conducted ablation experiments on the WIKIWHY dataset.

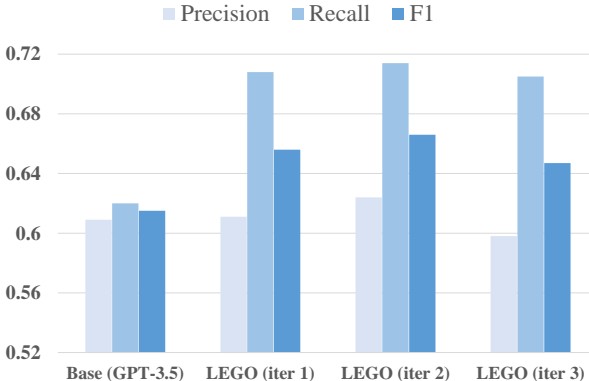

Figure 9: Performance of our framework with knowledge integration module (one round of interaction).

## 3.5 Discussion and Analysis

**Impact of Knowledge Integration Module**  Our ablation experiments with GPT-3.5 as base LM are presented in Table 9. 1) *Without knowledge integration module*, we plot the performance comparison of LEGO without the knowledge integration module in Figure 8. We can observe that the precision of the model experiences a continuous decline in the iterative process. This indicates that without fine-grained knowledge support, relying solely on iterative self-feedback is insufficient to effectively improve the generated explanations. 2) *With knowledge integration module (one round of interaction)*, as shown in Figure 9, the precision of LEGO gradually increases demonstrates that this module provides the necessary fine-grained knowledge. 3) *With knowledge integration module (multiple round of interaction)*, as illustrated in Table 9, in this setting, the performance of the model does not further improve. This could be attributed to the fine-grained knowledge stored in Observations is sufficient after one round of interaction. The more times the two analysts interact, the more noisy the collected knowledge may become, resulting in inefficient information in the feedback. Furthermore, Based on our analysis, the knowledge required for explanations is often fine-grained, such as "The fleet of the seventh voyage of the Ming Treasure Voyages." Firstly, this type of question cannot be directly answered through an API; it needs to be broken down into components like "Ming Treasure Voyages" and "the seventh voyage." Additionally, the information related to "the seventh voyage" is located on line 370 of the search page for "Ming Treasure Voyages," making knowledge localization a challenge. Moreover, as depicted in Figure 2,

the thinking process can accumulate a significant amount of information. Relying solely on the memory of a single agent can lead to the omission of crucial information during reasoning. Our approach of using multiple agents effectively mitigates this issue, as demonstrated.

**Impact of Iterative Feedback Module**  This module aims to provide multi-aspect feedback during the iterative process, allowing the Explainer to supplement fine-grained knowledge and task-specific commonsense knowledge. Take Figure 9 as an example, we observe a significant improvement in Recall (8.8%) after one round of refinement, we attribute this improvement to the incorporation of commonsense knowledge into the explanations. However, after one iteration, the improvement of Recall slows down or even decreases, which may be the modification of commonsense knowledge in the subsequent iterations is not obvious and multiple iterations will increase the length of explanation and lead to the decrease in precision.

**Refinement ability of LLM**  By comparing the recall of ordered evaluation of our approaches based on different underlying models (Instruct-GPT and GPT-3.5), we observed that InstructGPT demonstrated a growth of 3.5%, while GPT-3.5 exhibited an increase of 9.4% after one round of iteration. We attribute this noticeable difference to the stronger task comprehension and commonsense induction abilities of GPT-3.5. Furthermore, we noticed that InstructGPT is more "conservative" during the iteration process as presented in Figure 10. Due to space limitations, we present the explanations generated by GPT-3.5 in Figure 11 in Appendix. We can find that the explanations of InstructGPT remained almost identical, with only a few words being replaced by synonyms. In contrast, the explanations generated by GPT-3.5 exhibited greater richness and diversity.

## 4  Related Work

**Causality Explanation Generation**  Understanding causality is one of the most central cognitive abilities of human beings, different from the causality identification (Caselli and Vossen, 2017; Zuo et al., 2021; Cao et al., 2021; Tran Phu and Nguyen, 2021) which can only distinguish whether there is a causal relationship, causality explanation generation is especially worth explore since it not only test if a model "knows" that causality but also if it "understands" the underlying mechanics of

**Cause**: similar nature of the railways and to foster cooperation and volunteer exchanges
**Effect**: Lynton & Barnstaple signing a twinning agreement with the Walhalla Goldfields Railway
**Explanation**: Railways with similar goals improve their chance of success by creating a joint venture.

-------------------------------------------------------------------------

**GPT-3 based explanation**

**Iteration 0**: Lynton & Barnstaple signed a twinning agreement with the Walhalla Goldfields Railway in order to foster cooperation and volunteer exchanges due to the similar nature of the railways.

**Iteration 1**: Lynton & Barnstaple signed a twinning agreement with the Walhalla Goldfields Railway to promote cooperation and exchange of ideas between the two similar railways.

**Iteration 2**: Lynton & Barnstaple signed a twinning agreement with the Walhalla Goldfields Railway to foster cooperation and exchange of ideas between the two similar railways.

Figure 10: The explanations generated by our method with InstructGPT as the base model.

why that is the case. Du et al. (2022) proposed a human-annotated explainable CAusal REasoning dataset (e-CARE) to explore, which contains over 21K causal reasoning questions, together with natural language formed explanations of the causal questions. As large language models (LLMs) grow larger and more sophisticated, Ho et al. (2023) introduced WIKIWHY, which built around a novel auxiliary task: explaining why an answer is true in natural language, while InstructGPT basedlines achieve only low human-evaluated correctness in the end-to-end answer and explain condition.

**Communicative Agents**   Large language models have exhibited remarkable multi-dimensional capabilities in tackling complex tasks. However, their effectiveness greatly hinges on human guidance to steer the conversation, a process that can present challenges and consume significant time (Brown et al., 2020; Yao et al., 2022; Yang et al., 2023). It is important to consider how to autonomously guide the communicative agents toward task completion while maintaining consistency with human intentions. Communication between AI agents can occur in a competitive setting or a cooperative setting (Hadfield-Menell et al., 2016; Silver et al., 2017; Bard et al., 2020; Dafoe et al., 2020; Meng et al., 2023; Kwan et al., 2023). Cooperative AI systems consider the requirements and capacities of other agents within the system, actively striving to collaborate and synchronize their actions. This approach offers numerous advantages, such as enhanced efficiency, improved decision-making,

and the ability to address intricate problems that surpass the capabilities of individual agents. Li et al. (2023) enables two communicative agents to engage in a conversation and cooperate with each other to solve assigned tasks. However, designing effective cooperative AI systems is still an active area of research, as it requires addressing a range of technical, ethical, and social challenges.

**Learning from Feedback**   The utilization of natural language feedback, generated by both humans and machines, has proven to be effective across various tasks. Reinforcement learning (RL) approaches have been employed to optimize for human preferences or task accuracy, resulting in the generation of valuable feedback. (Bai et al., 2022a; Lu et al., 2022; Le et al., 2022). Recently LLMs have been used to generate feedback for a general domain solution (Yang et al., 2022; Fu et al., 2023; Peng et al., 2023). To make better use of feedback, pairs of feedback and revision have been employed to learn supervised refiners and correctors (Schick et al., 2022; Bai et al., 2022b; Yasunaga and Liang, 2020). However, gathering supervised data from humans is costly, to overcome this, Welleck et al. (2022) selected the best output by relying on knowing the ground truth at test time. Madaan et al. (2023) proposes soliciting feedback from an LLM on its own output, refining the output with feedback, and repeating this feedback-refine process.

## 5   Conclusion

We introduce LEGO, a Multi-agent Collaborative Framework with Role-playing and Iterative Feedback for causality explanation generation. Specifically, we treat LLM as character malleable LEGO block and utilize role-playing to assign specific roles to five LLMs, i.e. Cause Analyst, Effect Analyst, Knowledge Master, Critic and Explainer. We devise a Fine-grained World Knowledge Integration Module to augment information about tasks for alleviating the phenomenon of spurious causal associations. Then, we leverage an Iterative Feedback and Refinement Module to improve the generated explanation by multi-aspect feedback. Extensive experiments on WIKIWHY and e-CARE show the superiority of our multi-agent framework in terms of causality explanation generation.

## Limitations

Our method aims to explore the cooperation of multi-agents on causality explanation. In the exper-

iments, we found that GPT-3.5 is more powerful than InstructGPT in terms of task comprehension and commonsense induction abilities, but due to the high cost of API calls, we did not further test the performance of GPT-4 in this complex task. In addition, Explanations come in various structures, as presented in the typology defined by Neves Ribeiro et al. (2022), multiple explanations may be valid, the datasets we based on covered a large proportion of explanations to simple 'why' questions which minimizes the variability of explanations to some extent. The experimental results show that the explanations generated by LLMs are not ideal in the ordered evaluation. Moreover, although the automatic evaluation results show the effectiveness of our method, 44.8% of the explanations are still judged by human to be worse than the gold reference. In future work, we intend to delve deeper into the underlying structure of causal explanations within LLMs.

## Acknowledgements

This work is supported by the National Key Research and Development Program of China (No. 2020AAA0106400), the National Natural Science Foundation of China (No. 62176257, 61976211). This work is also supported by the Strategic Priority Research Program of Chinese Academy of Sciences (Grant No.XDA27020100 ), the Youth Innovation Promotion Association CAS, and Yunnan Provincial Major Science and Technology Special Plan Projects (No.202202AD080004).

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

# A   Dateset Details

## A.1   WIKIWHY

WIKIWHY (Ho et al., 2023) is a large-scale QA dataset built around explaining why an answer is true in natural language, which contains over 9,000 "why" question-answer-rationale triples, grounded on Wikipedia facts across a diverse set of topics. Each entry contains a cause-effect pair and rationale explaining the pair's causal relation. On average, each rationale contains 1.5137 elements. The statistics for the reasoning component are shown in Table 4.

## A.2   e-CARE

e-CARE (Du et al., 2022) is a large human-annotated explainable causal reasoning dataset,

| WikiWhy Statistics | |
|---|---|
| # of Train | 7397 |
| # of Dev | 1004 |
| # of Test | 1005 |
| # of Rationale | 9406 |
| # of Rationale Elements | 14238 |
| Avg. # Rationale Length | 1.5137 |
| Avg. # Tokens per Element | 16.697 |

Table 4: WikiWhy Summary Statistics

which contains over 21K causal reasoning questions, together with natural language formed explanations of the causal questions, The statistics are shown in Table 5.

| e-CARE Statistics | | |
|---|---|---|
| | Causal Questions | Uniq. Explanations |
| Train | 14928 | 10491 |
| Dev | 2132 | 2102 |
| Test | 4264 | 3814 |
| Total | 21324 | 13048 |

Table 5: Corpus level statistics of the e-CARE dataset. Uniq. Explanations refer to the explanations that only correspond to a single causal fact.

## A.3   Examples from Datasets

Table 6 contains examples from WikiWhy and e-CARE. $c$ denotes cause, $e$ effect, and $x$ is the explanation of the cause-effect pair.

# B   Evaluation Details

## B.1   Metrics for WIKIWHY

While the still developing area of text generation has measures and proxies for similarity that help with simple sequences, comparing reasoning sequences or rationale sets requires more involved measures. Ho et al. (2023) proposed two related metrics, unordered and ordered, to handle sets and sequences, respectively.

**Unordered Evaluation**   This first approach compares the ideas contained in the predictions and references. First, split predicted and reference explanations into "ideas" or "steps" by sentence. Then compute a matrix of pairwise similarity scores before using a threshold to classify "matches". Since

| WIKIWHY | Example |
| --- | --- |
| $c$ | The thermal stress at dawn and dusk. |
| $e$ | The boulders on Ceres are brittle and degrade rapidly. |
| $x$ | The thermal temperatures change so drastically the rocks expand and contract. This process weakens the structural integrity of the rocks. |
| $c$ | The duration of Hotel California was longer than songs generally played by radio stations. |
| $e$ | Don Felder had doubts about the 1997 Eagles song Hotel California. |
| $x$ | Most songs are only 3-4 minutes long. Hotel California is over 6 minutes. People would not want to listen to same song on radio for that long. |
| $c$ | Seeing the Castle of Cagliostro entrenched in Yamazaki that Japan can make high-quality films. |
| $e$ | Director Takashi Yamazaki modeled his 2019 film Lupin III: The First after The Castle of Cagliostro. |
| $x$ | Viewing The Castle of Cagliostro inspired Takashi Yamazaki. Out of national pride, Takashi Yamazaki followed a model that he believed would produce quality films. |
| $c$ | The geographic isolation of the Hupa homeland. |
| $e$ | The Hupa had few interactions with early European explorers up to the 19th century. |
| $x$ | The Hupa's homeland was separated by bodies of water or mountains. Not many people could get to the Hupa's homeland. |
| $c$ | The use of coal power in Turkey. |
| $e$ | Burning coal leads to air pollution. Air pollution causes sickness and early death. Sick and dead people cannot work. |
| $x$ | 1.4 million working days were lost across the population of Turkey in 2019. |

| e-CARE | Example |
| --- | --- |
| $c$ | He was infected with gram-positive bacteria. |
| $e$ | The doctor raised the lysozyme in his body. |
| $x$ | Lysozyme destroys cell wall of bacteria. |
| $c$ | The researcher investigated the premature death in these pet birds. |
| $e$ | He found they all died of Malnourishment. |
| $x$ | Malnourishment is a leading cause of premature death in pet birds. |
| $c$ | It is quite cold here in winter, and the temperature can reach as low as minus 30 degrees. |
| $e$ | In winter here, people wear clothes with very good thermal insulation when they go out. |
| $x$ | Clothing provides protection from the elements by increasing the insulating capacity of the body. |
| $c$ | Mary sent a emoticon c̈ryingẗo her boyfriend on her cell phone. |
| $e$ | Her boyfriend immediately called to comfort her. |
| $x$ | Emoticons are combinations of characters used to represent various emotions. |

Table 6: Examples from datasets

a single prediction sentence may contain multiple reference ideas, so keep separate counts of precise prediction steps and covered reference steps. These counts are then micro-averaged for the test set's overall precision, recall, and F1 scores.

**Ordered Evaluation** To respect the structure of multi-hop explanations, the method penalizes incorrectly ordered explanations. Here, use the previously generated pairwise score matrix and its alignments to generate all possible assignments of prediction sequence elements to reference elements. Then compute the length of the longest common subsequence (LCS) between a prediction alignment against the reference labels for each candidate assignment. This length becomes the count of correctly incorporated structural elements– true positives. Note that under this scheme, the repeated ideas in the prediction are discounted by the LCS-style alignment process.

We employ the BERTScore metric to measure sentence similarity (not evaluated directly using BERTScore). Taking the unordered evaluation algorithm proposed by (Ho et al., 2023) as an example, Precision and Recall are obtained using the following formulas:

$$Precision = \frac{precise}{prediction}, Recall = \frac{covered}{relevant}$$

Where "predictions" represents the number of sentences in the generated explanations, "relevant" denotes the number of sentences in the reference explanations, "**precise**" indicates how many of the generated sentences correspond to sentences in the explanations, and "**covered**" represents the number of sentences in the reference explanations that were successfully predicted.

For example, consider the following examples of reference explanations and generated explanations:

**Reference Explanation**: Opening the highway brought in an influx of unsafe people. With the higher traffic from a highway it would be hard to police the unsafe people. With them being harder to police it would become safer for them and they could drug deal and prostitute in the open.

**Generated Explanation**: When Interstate 5 was opened, it diverted traffic away from these areas and caused a decline in economic activity. With less commerce and people around, it became easier for illegal activities and establishments to operate without attracting attention from law enforcement or regular citizens.

**Score**: precise=2, predictions=2, covered=3, relevant=3.

The proposed algorithm keeps **separate counts of precise prediction steps and covered reference steps**, which effectively addresses the situation where **a single prediction sentence may contain multiple reference ideas**. For further details of the algorithms, please refer to the evaluation methods provided by (Ho et al., 2023).

For a fair comparison, we follow Ho et al. (2023) to select BERTScore using a large DeBERTa (He et al., 2020) model (microsoft/deberta-xlarge-mnli)[2] at a threshold of 0.64.

### B.2 Metric for e-CARE

**Causal Explanation Quality (CEQ)** Du et al. (2022) proposed a novel causal explanation quality evaluation metric (namely, CEQ score) as a step towards directly measuring the quality of generated explanations. Specifically, let $C$, $E$ and $X$ denote the cause, the effect and the generated explanation, respectively. Formally, the CEQ score is defined:

$$CEQ = \delta_{cs} = cs(C, E|X) - cs(C, E)$$

where $cs(C, E)$ is the original causal strength between $C$ and $E$; $cs(C, E|X)$ is the causal strength after involvement of the additional explanation information. The explanation enhanced causal strength $cs(C, E|X)$ is defined as:

$$cs(C, E|X) = max[cs(C + X, E), cs(c, E + X)]$$

where "+" denotes the string concatenate operation. Therefore, the CEQ score is positively related to the increase of causal strength between C and E after the involvement of the explanation X.

we employ a widely-adopted model-agnostic method proposed by Luo et al. (2016) to calculate the causal strength. The model-agnostic nature enable us to avoid reliance on certain models and keep the fairness of evaluation. Specifically, the phrase-level causal strength is derived through synthesizing the word-level causality.

$$cs(C_A, E_B) = \frac{1}{N_{C_A} + N_{E_B}} \sum_{w_i \in C_A, w_j \in E_B} cs(w_i, w_j)$$

where $(C_A, E_B)$ is an arbitrary causal fact; $N_{C_A}$ and $N_{E_B}$ are the number of words within $C_A$ and $E_B$, respectively; $cs(w_i, w_j)$ is the causal strength

---

[2] https://huggingface.co/microsoft/deberta-xlarge-mnli

between word $w_i$ and $w_j$, which is estimated from a large corpus as:

$$cs(w_i, w_j) = \frac{Count(w_i, w_j)}{Count(w_i)Count(w_j)^\alpha}$$

where $\alpha$ is a penalty coefficient and empirically set $\alpha = 0,66$ as same as Luo et al. (2016). We use the average CEQ score as the final score.

### B.3 Human Evaluation

Automatic metrics may not reliably evaluate results produced by models with few-shot capabilities like GPT-3. In light of this, we select the highest scoring explanations for each set of experiments for additional fine-grained evaluation and measure the agreement among the annotators.

(1) For each human evaluation task, we present a panel of three undergraduate students a random sample of 50 entries from each setting and the following binary True/False criteria guidelines:

- **Correctness**: Mark true if and only if the explanation is both complete and satisfying.

- **Concision**: Mark true if the explanation says everything it needs to say and nothing more. Mark false if extra information is included.

- **Fluency**: Is the explanation writing fluent? Mark false if there are any mechanical mistakes.

- **Validity**: Does the explanation make logical sense? Ignore whether or not the explanation successfully explains the cause/effect relation. Mark false if the explanation contains any illogical or untrue conclusions.

- **Win/Tie/Lose**: Compare the generated explanation against the provided reference. Mark Win if you prefer the generated explanation, Tie if you have no preference, and Lose if you prefer the reference explanation.

(2) Using the explanations of LEGO (GPT-3.5) as references, we compared different baselines against LEGO (GPT-3.5) separately. This was done to visually demonstrate the superiority of LEGO (GPT-3.5), as shown in Table 8.

(3) We measured the agreement among three annotators, and the table below displays our **Fleiss' Kappa** scores for the annotation agreements of Win/Tie/Lose (Compare the generated explanation against the provided reference).

| Model | K |
|---|---|
| GPT-3.5 | 0.84 |
| ReAct | 0.79 |
| LEGO (InstructGPT) | 0.76 |
| LEGO (GPT-3.5) | 0.78 |

Table 7: The agreement among three annotators

**Cause**: similar nature of the railways and to foster cooperation and volunteer exchanges
**Effect**: Lynton & Barnstaple signing a twinning agreement with the Walhalla Goldfields Railway
**Explanation**: Railways with similar goals improve their chance of success by creating a joint venture.
-----------------------------------------------------------------------------------
**GPT-3.5 based explanation**

**Iteration 0**: Lynton & Barnstaple signed a twinning agreement with the Walhalla Goldfields Railway in order to foster cooperation and volunteer exchanges due to the similar nature of the railways.

**Iteration 1**: The similar nature of the railways, as well as the desire to foster cooperation and volunteer exchanges, led Lynton & Barnstaple to sign a twinning agreement with the Walhalla Goldfields Railway..

**Iteration 2**: Organizations often form partnerships or twinning agreements with similar groups or organizations in order to enhance collaboration and mutual exchange. The Lynton & Barnstaple and Walhalla Goldfields railways signed a twinning agreement due to their similar nature and desire to foster cooperation and volunteer exchanges.

Figure 11: The explanations generated by our method with GPT-3.5 as the base model.

## C Details of Ablation Experiments

As shown in Table 9, we conduct extensive ablation experiments on WIKIWHY dataset.

## D Case Analysis

The cases we show are in Figure 12 and 13

## E Prompts

We include all the prompts used in the experiments in Figures 14-17.

**Cause**: The fleet of the seventh voyage of the Ming Treasure Voyages was sailing under favorable conditions and running before the southwest monsoon on their homeward voyage.
**Effect**: The fleet of the seventh voyage of the Ming Treasure Voyages did not stop at Ceylon or southern India.

**Initial Explanation (GPT-3.5)**: The fleet was sailing under favorable conditions and running before the southwest monsoon on their homeward voyage. This means that they were trying to make good time to get back home and did not want to make any unnecessary stops that would slow them down.

**Observation**:

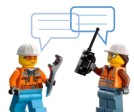

The fleet of the seventh voyage of the Ming Treasure Voyages: The fleet of the seventh Ming Treasure Voyage set sail in the early summer of 1430, taking advantage of the favorable southwest monsoon …
Ceylon, southern India and southwest monsoon: Ceylon, also known as Sri Lanka, is an island country located in the Indian Ocean, off the southern coast of India. It experiences a tropical climate and is greatly influenced by the southwest monsoon …

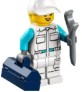

**Observation Feedback**: The Explanation is not a simple concatenation of Cause and Effect, but ignores that the southwest monsoon typically arrives in Ceylon around May or June and lasts until September.
**Commonsense Feedback**: Fleets need to pay close attention to the weather forecast and marine meteorological information, try to avoid being involved in the storm.

**LEGO Explanation**: The monsoon season typically arrives in Ceylon around May or June and lasts until September, which could have created risky sailing conditions for the fleet. Therefore, the fleet did not stop at these places to avoid being involved in the storm and continued their journey to get back home as quickly and safely as possible.

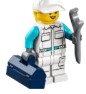

**Observation Feedback**: None.
**Commonsense Feedback**: Sailors and ships tend to avoid dangerous weather conditions in order to minimize the risk of damage to the ship and loss of life.

**Gold Explanation**: A monsoon was striking Ceylon or southern India at the time and the fleet of the seventh voyage of the Ming Treasure Voyage did not want to be caught in the storm.

Figure 12: Case study.

**Cause**: The Irish Republican Army had not resumed its ceasefire or agreed to disarm.

**Effect**: Sinn Fein had been elected to take part in the multi-party negotiations in Belfast for the 1996 Manchester bombing but were barred.

**Initial Explanation (GPT-3.5)**: Sinn Fein was elected to take part in the negotiations because they were considered a key player in Irish politics at the time. However, they were barred because the Irish Republican Army had not resumed its ceasefire or agreed to disarm.

**Observation**: 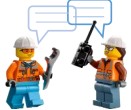

The Irish Republican Army, What is the significance of ceasefire and disarmament in the context of the conflict in Northern Ireland?: …Ceasefires allowed the two sides to engage in political negotiations and attempt to find a long-term solution to the conflict. Disarmament was seen as an essential part of the peace process, as it removed the potential for violent acts and reduced the potential for future conflict… What was Sinn Fein's position on ceasefire and disarmament during the time of the 1996 Manchester bombing? : At the time of the 1996 Manchester bombing, Sinn Fein's position on ceasefire and disarmament was that they wished for an end to violence on all sides but would only support disarmament once they were satisfied that the British government was committed to addressing the underlying causes of the conflict …

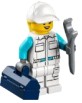

**Observation Feedback**: The current Explanation is a concatenation of Cause and Effect content. The explanation in the Task needs to be modified to reflect the fact that Sinn Fein was barred from the negotiations due to the IRA's failure to resume a ceasefire or agree to disarm, rather than their mere association with the IRA.

**Commonsense Feedback**: Political negotiations require trust and security. A ceasefire and disarmament are often prerequisites for progress in peace talks.

**LEGO Explanation**: Sinn Fein was elected to take part in the negotiations due to their importance in Irish politics. However, they were ultimately barred from the negotiations because the IRA had not resumed its ceasefire or agreed to disarm, which was necessary for progress in peace talks. Political negotiations require trust and security for progress, and a ceasefire and disarmament are often prerequisites.

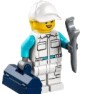

**Observation Feedback**: The Explanation is not a simple concatenation of Cause and Effect, which is correct.

**Commonsense Feedback**: Political negotiations require trust and security for progress, and a ceasefire and disarmament are often prerequisites. The IRA's involvement in peace talks was crucial to the success of the process, and failure to disarm was a significant obstacle.

**Gold Explanation**: For negotiations to occur there needs to be a ceasefire or a disarming for safety. Since the Irish Republican Army did not resume the ceasefire or disarm it was unsafe to take part in the negotiations so they were barred.

Figure 13: Case study.

**Cause**: There were time constraints to writing "Boruto: Naruto the Movie"
**Effect**: Hiroyuki Yamashita felt pressured writing "Boruto: Naruto the Movie"
**Explanation**: Creativity is difficult when put on a strict timetable. There was a need to both produce a good movie and do so on a strict time budget. These two demands put stress on Hiroyuki Yamashita while he worked.

**Cause** : Homer P. Rainey had liberal views.
**Effect** : Homer P. Rainey was fired by the University of Texas in 1944.
**Explanation** : If the University of Texas is conservative, they wouldn't want people working there who have liberal views.

**Cause** : the large size and reddish tint of red maple buds
**Effect** : Red maple buds which form in fall and winter are often visible from a distance.
**Explanation** : The color red stands out from a distance, so if the buds are red in the fall and winter, you'd be able to see them from a distance.

**Cause** : There were advances in technology, lower energy prices, a favorable exchange rate of the United States dollar, and lower alumina prices.
**Effect** : Productions costs of aluminum changed in the late 20th century.
**Explanation** : With advances in technology, prices of manufacturing change usually because they are now easier and cheaper to make. In this case it is aluminum that the price changed on because the technology improved the process.

Figure 14: Initial generation prompt for WIKIWHY

**Cause**: The child ran towards hippos.
**Effect**: His parents stopped him.
**Explanation**: Hippos are among the megafaunal species dangerous to humans.

**Cause** : Jack keeps the goats alone.
**Effect** : The goat has a poor appetite.
**Explanation** : Goats are herd animals so depend on the companionship of other goats.

**Cause** : The photographer has to edit digital images.
**Effect** : He understood the pixels as their basic building blocks.
**Explanation** : Pixels are the basic building blocks of all digital images.

**Cause** : The researcher separated integral membrane proteins from the tonoplast.
**Effect** : They are TIPs.
**Explanation** : TIPs are integral membrane proteins in the tonoplast.

**Cause** : Tom eats a lot of eggs every day.
**Effect** : The cholesterol content in his body is extremely high, so he suffers from gallstones.
**Explanation** : Gallstones are lumps composed mainly of cholesterol.

**Cause** : The shamans were to start their sacred ceremonies.
**Effect** : They moved to a cave.
**Explanation** : Caves are sacred ceremonies sites utilized by the shamans.

Figure 15: Initial generation prompt for e-CARE

**Observation**: Why Orville E. Babcock was unpopular: Orville E. Babcock (1835-1884) was a U.S. Army officer and personal secretary to President Ulysses S. Grant. Babcock was ... ; Why President Grant of the United States dismiss Orville E. Babcock: President Ulysses S. Grant dismissed his personal secretary, Orville E. Babcock, in 1876 due to allegations of his involvement in a major corruption scandal known as the Whiskey Ring.The Whiskey Ring ...

**Task**: Cause: Orville E. Babcock was unpopular. Effect: In 1876, President Grant of the United States dismissed Orville E. Babcock from the White House under public pressure.

**Explanation**: Orville E. Babcock's unpopularity led to public pressure on President Grant to remove him from the White House.

Check whether the Explanation is just a concatenation of Cause and Effect content, if so, please point out and give feedback based on the information in the Observation and Task.
What common sense knowledge is required to explain the causal relationship in the task?
If the current explanation is perfect, return None.

**Observation Feedback**: The Explanation is a simple concatenation of Cause and Effect, which is incorrect. Based on Observation, Orville E. Babcock was unpopular due to his involvement in the Whiskey Ring scandal.

**Commonsense Feedback**: Politicians dismiss unpopular figures around them. Politicians typically try to remain popular with the people.

Figure 16: Feedback prompt of Critic LLM (truncated).

**Task**: Cause: Orville E. Babcock was unpopular. Effect: In 1876, President Grant of the United States dismissed Orville E. Babcock from the White House under public pressure.

**Explanation**: Orville E. Babcock's unpopularity resulted in public pressure on President Grant, which led to his dismissal.

Check whether the Explanation is just a concatenation of Cause and Effect content, if so, please point out and give feedback based on the information in the Observation and Task.

**Observation Feedback**: The Explanation is a simple concatenation of Cause and Effect, which is incorrect. Based on Observation, Orville E. Babcock was unpopular due to his involvement in the Whiskey Ring scandal.
What common sense knowledge is required to explain the causality in the task?

**Commonsense Feedback**: Politicians dismiss unpopular figures around them. Politicians typically try to remain popular with the people.

Okay, impove the Explanation using these feedback (Please keep Explanations concise rather than redundant):
**Explanation**: Politicians dismiss unpopular figures around them to remain popular with the people. Orville E. Babcock's scandal resulted in public pressure on President Grant, which led to his dismissal.

Check whether the Explanation is just a concatenation of Cause and Effect content, if so, please point out and give feedback based on the information in the Observation and Task.

**Observation Feedback**: None
What common sense knowledge is required to explain the causality in the task?
**Commonsense Feedback**: None

Figure 17: Refine prompt of Explainer LLM (truncated).

Never forget you are a Cause Analyst and I am a Effect Analyst. Never flip roles!

We share a common interest in collaborating to complete a task that reasoning about the given Cause-Effect pair.

I must help you to complete the task.

You need to reason ONLY in the following two ways:

1. Thought with necessary Ask:
Thought: <YOUR_THOUGHT>
Ask: <YOUR_QUESTION>

2. Thought without any Ask:
Thought: <YOUR_THOUGHT>
Ask: None

The "Thought" describes your reasoning process which includes entities or questions that need to ask knowledge base , and always starts with "I need to analyze from the Cause. Related entities in Cause are:". The paired "Ask" presents the entities or questions in the Thought.

Here are some examples:
Task：Cause: Orville E. Babcock was unpopular. Effect: In 1876, President Grant of the United States dismissed Orville E. Babcock from the White House under public pressure.
Thought: I need to analyze from the Cause. Related entities in Cause are:[Orville E. Babcock]. So I need to ask about Orville E. Babcock, find why he was unpopular.
Ask: Orville E. Babcock, why Orville E. Babcock was unpopular
…

You must give me ONLY one Thought at a time!
Never forget you are Cause Analyst !
Do not output anything else other than Thought and the optional corresponding Ask!
Here is the task: <TASK>. Never forget our task!
Your Thought must prioritize what is mentioned in the Task.
Now you must start to reason using the two ways described above.

Figure 18: Inception prompt of Cause Analyst LLM (truncated).

Never forget you are an Effect Analyst and I am a Cause Analyst. Never flip roles!

We share a common interest in collaborating to complete a task that reasoning about the given Cause-Effect pair.

You must help me to complete the task.

You need to reason ONLY in the following way:

1. Thought with necessary Ask:
Thought: <YOUR_THOUGHT>
Ask: <YOUR_QUESTION>

2. Thought without any Ask:
Thought: <YOUR_THOUGHT>
Ask: None

The "Thought" describes your reasoning process which includes entities or questions that need to ask knowledge base , and always start with "I need to analyze from the Effect. Related entities in Effect are:". The paired "Ask" presents the entities or questions in the Thought.

Here are some examples:

Task：Cause: Orville E. Babcock was unpopular. Effect: In 1876, President Grant of the United States dismissed Orville E. Babcock from the White House under public pressure.

Thought: I need to analyze from the Effect. Related entities in Effect are:[President Grant of the United States, Orville E. Babcock]. As stated in the main reasoning process, Orville E. Babcock was unpopular due to his involvement in the Whiskey Ring scandal and Babcock was a personal secretary and close confidant of President Ulysses S. Grant. So President Ulysses S. Grant may be criticized by the public and may cause his approval rating to drop. I need to find more information about why President Grant of the United States dismiss Orville E. Babcock.

Ask: why President Grant of the United States dismiss Orville E. Babcock

…

You must give me one Thought at a time!

Never forget you are an Effect Analyst !

Do not add anything else other than your Thought and the optional corresponding Ask!

Here is the task: <TASK>. Never forget our task!

Your Thought must prioritize what is mentioned in the Task.

Now you must start to reason using the two ways described above.

Figure 19: Inception prompt of Effect Analyst LLM (truncated).

| Setting | Fine Grained Human Evaluation | | | | | | |
|---|---|---|---|---|---|---|---|
| | Correctness | Concision | Fluency | Validity | Win (↑) | Tie | Lose (↓) |
| GPT-2 vs LEGO | 0.113 | 0.593 | 0.146 | 0.100 | 0.073 | 0.086 | 0.841 |
| GPT-3 vs LEGO | 0.226 | 0.433 | 0.253 | 0.213 | 0.133 | 0.333 | 0.534 |
| GPT-3.5 vs LEGO | 0.293 | 0.406 | 0.326 | 0.280 | 0.206 | 0.340 | 0.454 |
| ReAct (GPT-3.5) vs LEGO | 0.306 | 0.393 | 0.313 | 0.326 | 0.213 | 0.346 | 0.441 |
| LEGO (InstructGPT) vs LEGO | 0.360 | 0.426 | 0.453 | 0.393 | 0.226 | 0.373 | 0.401 |

Table 8: Other baselines vs LEGO (GPT-3.5).

| Experiments | Unordered | | | Ordered | | |
|---|---|---|---|---|---|---|
| | Precision | Recall | F1 | Precision | Recall | F1 |
| LEGO | **0.624** | **0.714** | **0.666** | **0.464** | **0.634** | **0.536** |
| w/ single Analyst | 0.596 | 0.705 | 0.646 | 0.455 | 0.617 | 0.523 |
| w/o Knowledge Master | 0.578 | 0.698 | 0.632 | 0.447 | 0.598 | 0.511 |
| w/o Critic | 0.584 | 0.685 | 0.630 | 0.449 | 0.608 | 0.516 |
| w/o interaction | 0.593 | 0.703 | 0.643 | 0.451 | 0.609 | 0.518 |
| w interaction (2 rounds) | 0.619 | 0.701 | 0.657 | 0.448 | 0.625 | 0.521 |

Table 9: Ablation experiments. We use GPT-3.5 as the base model.