# OpenReview forum: "LEGO: A Multi-agent Collaborative Framework with Role-playing and Iterative Feedback for Causality Explanation Generation"
_EMNLP/2023/Conference — EMNLP 2023 Findings_

### Official Review · Reviewer_dYrk · 2023-08-03

**Soundness:** 4

**Excitement:**

4: Strong: This paper deepens the understanding of some phenomenon or lowers the barriers to an existing research direction.

**Paper Topic And Main Contributions:**

This paper proposes a multi-agent collaborative framework for causality explanation generation. The framework consists of five agents: cause analyst, effect analyst, knowledge master, explainer, and critic. The cause analyst and effect analyst conduct bi-directional reasoning between the cause and the effect while enquiring world knowledge from the knowledge master. After that, the explainer and critic agents generate the causality explanation via iterative feedback and refinement.

**Questions For The Authors:**

Did you measure the agreement among the annotators?

**Reasons To Accept:**

The proposed bi-directional reasoning mechanism in the cause analyst and effect analyst is novel and interesting. It is also helpful in mitigating the problem of spurious causal association in LLMs.

The proposed framework is well-motivated and the intuition behind each agent is clearly explained.

The experiments in this paper are comprehensive. The ablation studies analyze the impact of each major component in the framework. It conducts human evaluation using both absolute-scale metrics (Correctness, Fluency, Validity, etc.) and ranking-based metrics (win, lose, or tie).


**Reasons To Reject:**

The presentation of the evaluation metrics and experiment results should be improved. It is not clear how to compute the precision, recall, and BERT-f1 scores results in Table 1. The results of automatic evaluations and human evaluations should be reported in separate tables. Please check the detailed comments in Presentation Improvements.

**Reproducibility:**

4: Could mostly reproduce the results, but there may be some variation because of sample variance or minor variations in their interpretation of the protocol or method.

**Reviewer Confidence:**

4: Quite sure. I tried to check the important points carefully. It's unlikely, though conceivable, that I missed something that should affect my ratings.

**Typos Grammar Style And Presentation Improvements:**

In appendix B.1, it mentions that it uses BERTScore for similarity measurements. Therefore, it seems that all the precision, recall, and F1 scores in Table 1 are computed by BERT score. However, the heading in Table 1 is “Precision, Recall, BERT-f1”, which is quite confusing. Does it mean the precision and recall scores are not computed by BERTScore?
Moreover, the results of automatic evaluations and human evaluations should be reported in separate tables. This is because automatic evaluations are conducted on the whole test set, while human evaluations are only conducted on 50 random samples. Presenting both automatic and human evaluation results in the same table (e.g., Table 1) may cause confusion.

---

> ### Author Rebuttal · Authors · 2023-08-28
>
> We’d like to thank your detailed notes and feedback.
>
> ### Respone to Question
> >**Answer for Question 1**: Did you measure the agreement among the annotators?
>
> We apologize that we don’t give detail description about human evaluation. In order to ensure the consistency of the human evaluation with previous works, we follow Matthew Ho [1] to conduct human evaluations via spreadsheets. We randomly shuffled the golden reference and the generated candidates, we don't reveal which column is the golden reference and which is the generated candidate in the evaluation setup. We measured the agreement among three annotators, and the table below displays our **Fleiss' Kappa** scores for the annotation agreements of Win/Tie/Lose (Compare the generated explanation against the provided reference).
>
> | Model |  K |
> | -------- |:----:|
> | GPT-3.5 | 0.84 |
> | ReAct | 0.79 |
> | LEGO (InstructGPT)| 0.76 |
> | LEGO (GPT-3.5) |0.78 |
>
> It can be observed that all scores exceed 0.7, indicating that the annotation results achieved substantial agreement. We appreciate your suggestions, and we will add it in the revised version.
>
> ### Respone to Reject
>
> We apologize that we don’t give detail description about the **automated evaluation algorithm proposed by Matthew Ho [1]**. To clarify, we employed the BERTScore metric to measure sentence similarity, and "BERT-f1" is derived from both Precision and Recall. Taking the unordered evaluation algorithm as an example, Precision and Recall are obtained using the following formulas:
>
> $Precision=\frac{precise}{predictions}$,  $Recall=\frac{coverd}{relevant}$
>
> Where "predictions" represents the number of sentences in the generated explanations, "relevant" denotes the number of sentences in the reference explanations, "**precise**" indicates how many of the generated sentences correspond to sentences in the explanations, and "**covered**" represents the number of sentences in the reference explanations that were successfully predicted.
>
> For example, consider the following examples of reference explanations and generated explanations:
>
> **Reference Explanation**: Opening the highway brought in an influx of unsafe people. With the higher traffic from a highway it would be hard to police the unsafe people. With them being harder to police it would become safer for them and they could drug deal and prostitute in the open.
>
> **Generated Explanation**: When Interstate 5 was opened, it diverted traffic away from these areas and caused a decline in economic activity. With less commerce and people around, it became easier for illegal activities and establishments to operate without attracting attention from law enforcement or regular citizens.
>
> *Score*: precise=2, predictions=2, covered=3, relevant=3.
>
> The proposed algorithm keeps **separate counts of precise prediction steps and covered reference steps, which effectively addresses the situation where a single prediction sentence may contain multiple reference ideas** (Appendix B). For further details on the algorithm, please refer to the evaluation methods provided by Ho et al. We intend to make our code publicly available in the future to facilitate upcoming research efforts. We appreciate your feedback and will add detail information about metrics in the revised version.
>
> ### Other Additions
>
> We appreciate your feedback. We will add detail information about evaluation algorithm and report The results of automatic evaluations and human evaluations in separate tables in the revised version.
>
> ### References
>
> [1] Ho M, Sharma A, Chang J, et al. Wikiwhy: Answering and explaining cause-and-effect questions. ICLR 2023

---

### Official Review · Reviewer_3dKC · 2023-08-07

**Soundness:** 3

**Excitement:**

2: Mediocre: This paper makes marginal contributions (vs non-contemporaneous work), so I would rather not see it in the conference.

**Paper Topic And Main Contributions:**

The paper presents a role-playing framework with multiple LLMs for causality explanation generation. According to the presented evaluation, the method improves over single LLM baselines on two datasets.

**Questions For The Authors:**

A) How did you determine significance?

**Reasons To Accept:**

The idea to augment the input before generating the explanation is interesting.

**Reasons To Reject:**

- Multiple explanations are valid, while only a single ground-truth explanation is available in the evaluation dataset. This situation is not well captured by automated metrics and hence evaluation by a single metric (BERT-score) is not sufficient. In addition, the human evaluation guidelines are ambiguous "Correctness: Mark true if and only if the explanation is both complete and satisfying.". Correctness (the truth and nothing but the truth) is different from completeness (the whole truth) and far different from satisfying (which is subject to interpretation).
- The impact of the role-playing framework remains unclear. Unfortunately, section 4 sheds little light as the single-metric issues persists, differences are subtle and not always conclusive (e.g., while the authors state that precision gradually increases in Figure 7, whereas it actually decreases in iteration 3).

**Reproducibility:**

3: Could reproduce the results with some difficulty. The settings of parameters are underspecified or subjectively determined; the training/evaluation data are not widely available.

**Reviewer Confidence:**

3: Pretty sure, but there's a chance I missed something. Although I have a good feel for this area in general, I did not carefully check the paper's details, e.g., the math, experimental design, or novelty.

**Typos Grammar Style And Presentation Improvements:**

- Figure 6 shows differences, Figure 7 absolute values. Please unify.
- Table 7 is in the appendix and should be referred to as such.

---

> ### Author Rebuttal · Authors · 2023-08-28
>
> Thank you for your time and feedback.
> ### Respone to Question
> >**Answer for Question A**:  How did you determine significance?
>
> The task of causal explanation generation as well as its evaluation are both challenging. *To clarify, the datasets [1,2] and evaluation metrics mentioned in the paper are not the contributions of our study*. As they have been widely accepted by NLP community, and in order to maintain consistency with related works, we strictly adhered to the evaluation methods used in previous research for our experiments, including both automated and human evaluation. The results of *automated evaluation* are presented in Table 1 and Table 3. In comparison to the baselines of a single agent (including the updated baseline ReAct), our multi-agent framework effectively enhances the quality of generated explanations. Furthermore, we tested our framework with different underlying language models (InstructGPT, GPT-3.5), and observed an **overall improvement of 8.5% and 2.8%** respectively (based on automated ordered evaluation). Additionally, the results of fine-grained *human evaluation* are displayed in Table 2. Explanations generated by the LEGO framework with GPT-3.5 received a **recognition rate of 75.3%** from human annotators. In response to the concern raised by Reviewer 8kHz about the challenge of intuitively comparing diverse baselines, we have conducted *additional human evaluation*, which can provide a more intuitive reflection of the significance of our results.  Using the explanations of LEGO (GPT-3.5) as references, we compared different baselines against LEGO (GPT-3.5) separately. This was done to visually demonstrate the relative superiority of other baselines over LEGO (GPT-3.5).
>
> | Model|Correctness  | Concision|Fluency | Validity |Win |Tie|Lose |
> | -------- |:----:| :----:|:----: | :----: | :----: |:----: |:----: |
> | **GPT-2** vs LEGO (GPT-3.5) | 0.113 | 0.593 | 0.146 | 0.100 | 0.073 | 0.086 |0.841 |
> | **GPT-3** vs LEGO (GPT-3.5)  | 0.226 | 0.433 | 0.253 | 0.213 | 0.133 | 0.333 |0.534 |
> | **GPT-3.5** vs LEGO (GPT-3.5)  | 0.293 | 0.406 | 0.326 |0.280 | 0.206 | 0.340 | 0.454 |
> | **ReAct (GPT-3.5)** vs LEGO (GPT-3.5)  | 0.306 | 0.393 | 0.313 | 0.326 | 0.213 | 0.346 |0.441 |
> | **LEGO (InstructGPT)** vs LEGO (GPT-3.5)  | 0.360 | 0.426 | 0.453 | 0.393 | 0.226 | 0.373 |0.401 |
>
> The results indicate that explanations generated by LEGO (GPT-3.5) are considered superior to GPT-3.5 in **70.7%** of cases and outperform the strongest baseline, ReAct, in **69.4%** of cases.
>
> ### Respone to Reject
> >**A. About datasets and evaluation metrics**
>
> We agree that some complex causal relationships may have multiple explanations. To clarify, **these widely used datasets and evaluation metrics mentioned in the paper are not the contributions of our study**. We have provided a detailed description of the datasets and evaluation metrics in the appendix. Explanations come in various structures, as seen in the typology defined by Ribeiro et al [3]. It's not an easy task to perfectly organize all possible explanations. The WikiWhy dataset includes two types of explanations (step sequence and sets) are likely not comprehensive, as a novel effort in QA explanations and reasoning, they probably cover a large proportion of explanations to simple ‘why’ questions. This minimizes the variability of explanations to some extent, facilitating the fair assessment of complex reasoning capabilities of language models. Given the widespread acceptance of these two datasets and their corresponding evaluation metrics within the community, there is no reason for us to reject the utilization of these datasets and forgo our exploration of the causal explanation task.
>
> In the paper, we extensively elaborate on the metrics for automated evaluation, emphasizing that evaluation is not based solely on a single metric (Appendix B.1). For a comprehensive understanding, please refer to Reviewer 3. Specifically, the concepts of "completeness" and "satisfying" within the "Correctness" are not contradictory. Considering that a gold explanation involves multiple fine-grained knowledge, it's reasonable to require the generated explanation to exhibit "completeness." Simultaneously, to prevent the model from arbitrarily combining these fine-grained knowledge or generating excessive unrelated information, the demand for the generated explanation to be "satisfying" to humans is also justifiable. To ensure the validity of human evaluations, we enlisted three graduate students with a certain level of expertise to conduct independent assessments, guaranteeing the credibility of the human evaluation results.
>
> >**B. The impact of our framework**
>
> In Answer A, we illustrate the **overall impact** of the role-playing framework. For **the impact of different modules** within the framework, such as interactions/iterations, please refer to Answer 4 in our response to Reviewer 8kHz. Additionally, in Figure 7, we present the performance of our framework with knowledge integration module (one round of interaction). As stated in section 4, through iterative refinement, the explainer incorporates multi-aspect feedback from the critic, ensuring that the generated explanations encompass essential information (i.e., worldly knowledge and common sense), thereby enhancing precision. However, irrelevant information is also inevitably present in the feedback. Consequently, after multiple iterations, this irrelevant information contributes to redundancy and disorder within the explanations. As the evaluation algorithm penalizes instances of sentence redundancy and disorder in the explanations, this leads to a decline in accuracy after multiple iterations. Therefore, the findings from our ablation experiments indicate that the **optimal round of iterations** is 2.
>
> ### Reference
> [1] Ho M, Sharma A, Chang J, et al. Wikiwhy: Answering and explaining cause-and-effect questions. ICLR 2023
> [2] Du L, Ding X, Xiong K, et al. e-CARE: a new dataset for exploring explainable causal reasoning. ACL 2022
> [3] Ribeiro D, Wang S, Ma X, et al. Entailment tree explanations via iterative retrieval-generation reasoner. NAACL 2022

---

### Official Review · Reviewer_8kHz · 2023-08-11

**Soundness:** 3

**Excitement:**

4: Strong: This paper deepens the understanding of some phenomenon or lowers the barriers to an existing research direction.

**Paper Topic And Main Contributions:**

This paper introduces a novel framework to enhance the performance of LLMs in the Causality Explanation Generation task. LLMs assume five distinct roles, encompassing input processing, external knowledge retrieval from the Internet, and iterative answer refinement. LLMs in these roles communicate using a static protocol with specific prompts to collaboratively complete the explanation task. The method was evaluated on two datasets: WIKIWHY and e-CARE. The primary contribution is the multi-agent collaborative framework and the communication protocol, which breaks the primary causality explanation task into several sub-tasks.

**Questions For The Authors:**

A: Given just the input cause-effect pair, it's challenging to assess the accuracy of a causality explanation. External knowledge, such as the monsoon example provided, can heavily influence the explanation. Different external knowledge might validate other seemingly spurious explanations. For example, If different external knowledge is available, e.g. the fleet was behind their schedule for homeward voyage, then the spurious explanation given in the paper might be appropriate as well. To address this variability, would it be viable to augment the dataset with multiple explanations for a single cause-effect pair?

B: Why were different evaluation metrics chosen for the datasets? Could you provide comprehensive results?

C: What informed the decision to break critic feedback into observation and commonsense? Was this grounded in theory or based on preliminary findings such as prompt engineering?

D: Do you have any insights on the optimal number of interactions/iterations and potential termination criterias?

**Reasons To Accept:**

The paper is well written with clear presentations, complemented by motivating examples and illustrations.

The proposed role-playing and iterative feedback framework for LLM is innovative. Similar collaborative frameworks have potential applications beyond causality explanation.

The authors undertook comprehensive experiments on two renowned datasets using various base LLMs. Ablation experiments were astutely constructed to isolate the impacts of distinct modules.

Insights on the optimal number of interactions between LLM agents in collaborative tasks are beneficial for the research community.

**Reasons To Reject:**

Different evaluation metrics were used for the datasets without a defined rationale, which undermines the results.

The majority of the automatic evaluation metrics in the paper gauge similarity between predicted explanations and the reference standard. It seems to me that accurate knowledge retrieval is more important than making casual reasoning in those evaluations. Given that knowledge retrieval is primarily executed by the ReAct framework with Internet access, the paper might benefit from an added baseline to assess a single ReAct agent's performance on the explanation task.

The human evaluation section needs elaboration. Did the three annotators work independently or engage in panel discussions? How were the 50 samples presented: randomly or in clusters?

In the results section, line 379, the author asserts significance without performing statistical analysis. A 4% improvement, given the human evaluation method, indicates that only 2 out of 50 samples were labeled differently. Without supporting evidence, this difference could be attributed to noise or evaluation bias.

The presentation of the Ablation experiments is ambiguous. Section 4 only addresses how the number of iterations and interactions affect LEGO's performance. However, a crucial comparison between LEGO (with only the knowledge integration module), LEGO (with only the iterative feedback module), and GPT-3.5 is absent. While some details are in the appendix (Table 7), an in-depth discussion in the main text would be beneficial.

**Reproducibility:**

4: Could mostly reproduce the results, but there may be some variation because of sample variance or minor variations in their interpretation of the protocol or method.

**Reviewer Confidence:**

4: Quite sure. I tried to check the important points carefully. It's unlikely, though conceivable, that I missed something that should affect my ratings.

**Typos Grammar Style And Presentation Improvements:**

The terms "iteration" and "interaction" are used without clarification in the results section. I presume "iteration" pertains to Iterative Feedback and Refinement, and "interaction" relates to Fine-grained World Knowledge Integration. Clarifying this distinction would be helpful.

---

> ### Author Rebuttal · Authors · 2023-08-28
>
> Thanks for your careful and insightful reviews.
>
> ### Respone to Question
> >**Answer for Question 1**: Would it be viable to augment the dataset with multiple explanations for a single cause-effect pair?
>
> We think your insights into this task are highly valuable. Evaluating generated explanations indeed poses specific challenges. To clarify, all the data is **sourced from Wikipedia** articles, which means that the explanations in the dataset are **based on established world knowledge**. External knowledge may not form a correct causal explanation. We agree with your point about "the variability of explanations." However, as seen in the typology defined by Ribeiro et al [1], explanations come in various structures. It's not an easy task to perfectly organize all possible explanations. The WikiWhy dataset includes two types of explanations (step sequence and sets) which may not comprehensive, as a novel effort in QA explanations and reasoning, they probably cover a large proportion of explanations to **simple ‘why’ questions and widely accepted** by the NLP community. This minimizes the variability of explanations to some extent, facilitating the fair assessment of complex reasoning capabilities of language models. We appreciate your suggestions and expect to explore more intricate structures and variability in explanations for complex "why" questions in our future work.
>
> >**Answer for Question 2**: Why were different evaluation metrics chosen for the datasets? Could you provide comprehensive results?
>
> The e-CARE and WikiWhy datasets have distinct annotation standards [2,3]. In Table 6, we have presented numerous examples from both datasets. The explanations in e-CARE focus on elaborating the causal facts at a conceptual level, typically encompassing **only one conceptual sentence**. For instance, "Emoticons are combinations of characters used to represent various emotions." (example 4 of e-CARE). Therefore, to maintain consistency with previous works, we are following the evaluation metrics proposed in the e-CARE paper. In contrast, explanations in WikiWhy exhibit **two structures**: multi-hop step sequences and rationale sets, rendering them more instantiated. According to statistics, the average length of explanations in this dataset is 1.5 sentences (Table 4), with some extending up to 6 sentences. Additionally, there is a fixed order among the sentences. Consequently, the WikiWhy paper introduces both ordered and unordered evaluation to compare the ideas contained in the predictions and references. We apologize for not providing a detailed description of the chosen evaluation metrics. We will add it in the revised version.
>
> >**Answer for Question 3**: What informed the decision to break critic feedback into observation and commonsense? Was this grounded in theory or based on preliminary findings such as prompt engineering?
>
> We are based on **preliminary findings from error analysis and relevant research papers**. Firstly, according to our statistics, the most prominent errors are the lack of common-sense knowledge and repetitive causal relationships (accounting for a combined **54%**). Secondly, many research studies [4,5] have indicated the success of multi-aspect feedback. To sum up, we decide to break critic feedback into observation and commonsense.
>
> >**Answer for Question 4**: Do you have any insights on the optimal number of interactions/iterations and potential termination criterias?
>
> Firstly, increasing the number of interactions might allow the Critic to **observe more relevant information**. However, simultaneously, this increase in information could **introduce noise and redundancy**. We validated this conclusion through the experiments presented in Table 7. The results from experiments with two interaction rounds were better than those without any interactions, but they were inferior to the setup with only one interaction. The more information Critic observes, the longer the feedback it provides, leading to more redundancy within the explanations provided by the explainer. While this might **be beneficial for improving Recall** (increasing the number of correct sentences in the generated explanations), evaluation metrics penalize excessive sentences. Secondly, increasing the number of iterations can also lead to **excessive explanation lengths, harming precision**. Based on our analysis, the generated common-sense feedback does not exhibit significant changes during iterations (as illustrated in Figure 10). However, the observed feedback accumulates within explanations (much of which could be informational noise), resulting in a decline in precision.
>
> ### Respone to Reject
> >**A. The concerns about evaluation metrics.** (Please refer to Answer2)
>
> >**B. Baseline updates**
>
> We appreciate your suggestion on experiment. Following your suggestion, we have added this experiment with a single ReAct agent and the updated paper revision contains the results (Table 1).
>
> | Model     |  | unordered| | \ | |ordered| |
> | -------- |:----:| :----:|:----: | :----: | :----: |:----: |:----: |
> | GPT-3.5 | 0.609 | 0.620 | 0.615 | \ | **0.467** | 0.557 | 0.508 |
> | ReAct (GPT-3.5) | 0.597 | 0.669 | 0.631 | \ | 0.455 | 0.595 | 0.516 |
> | LEGO (GPT-3.5) | **0.624** | **0.714** | **0.666** | \ | 0.464 | **0.634** | **0.536** |
>
> Based on our analysis, the knowledge required for explanations is often fine-grained, such as "The fleet of the seventh voyage of the Ming Treasure Voyages." Firstly, this type of question **cannot be directly answered** through an API; it needs to be broken down into components like "Ming Treasure Voyages" and "the seventh voyage." Additionally, the information related to "the seventh voyage" is located on **line 370 of the search page** for "Ming Treasure Voyages," making knowledge localization a challenge. Moreover, as depicted in Figure 2, the thinking process can accumulate a significant amount of information. Relying solely on the memory of a single agent can lead to **the omission of crucial information** during reasoning. Our approach of using multiple agents effectively mitigates this issue, as demonstrated.
>
> >**C. About the human evaluation**
>
> To ensure consistency with prior works, we strictly adhere to the organizational approach of Ho et al [2]. The three annotators work **independently** on **randomly selected samples**. The human evaluations were conducted via spreadsheets. We randomly shuffled the golden reference and the generated candidates, we don't reveal which column is the golden reference and which is the generated candidate in the evaluation setup.
>
> >**D. Differences in human evaluation results**
>
> In fact, we had three annotators independently evaluate 50 randomly selected samples, with a maximum score of 150 for each metric. However, due to the evaluation metrics set by Ho et al, which are intended to count the number of samples in the current results that meet the “Correctness” metric, this **cannot serve as an intuitive measure** for comparing differences between different outcomes. We appreciate your suggestion and have conducted additional human evaluation. Using the explanations of **LEGO (GPT-3.5) as references**, we compared different baselines against LEGO (GPT-3.5) separately. This was done to visually demonstrate **the relative superiority of other baselines over LEGO (GPT-3.5)**.
>
> | Model|Correctness  | Concision|Fluency | Validity |Win |Tie|Lose |
> | -------- |:----:| :----:|:----: | :----: | :----: |:----: |:----: |
> | **GPT-2** vs LEGO (GPT-3.5) | 0.113 | 0.593 | 0.146 | 0.100 | 0.073 | 0.086 |0.841 |
> | **GPT-3** vs LEGO (GPT-3.5)  | 0.226 | 0.433 | 0.253 | 0.213 | 0.133 | 0.333 |0.534 |
> | **GPT-3.5** vs LEGO (GPT-3.5)  | 0.293 | 0.406 | 0.326 |0.280 | 0.206 | 0.340 | 0.454 |
> | **ReAct (GPT-3.5)** vs LEGO (GPT-3.5)  | 0.306 | 0.393 | 0.313 | 0.326 | 0.213 | 0.346 |0.441 |
> | **LEGO (InstructGPT)** vs LEGO (GPT-3.5)  | 0.360 | 0.426 | 0.453 | 0.393 | 0.226 | 0.373 |0.401 |
>
> >**E. About the Ablation experiments**
>
> Due to the limited space, in Section 4, we focus on the primary conclusions (impact of knowledge integration and impact of iterative feedback). We appreciate your suggestion on the suggestion. In the revised version, we will incorporate a comparison between LEGO (with only the knowledge integration module), LEGO (with only the iterative feedback module), and GPT-3.5.
>
> ### Other Additions
>
> As you have correctly understood, "interaction" relates to fine-grained world knowledge integration and "iteration" involves iterative feedback and refinement. We will clarify these terms in the revised version.
>
> ### Reference
> [1] Ribeiro D, Wang S, Ma X, et al. Entailment tree explanations via iterative retrieval-generation reasoner. NAACL 2022
>
> [2] Ho M, Sharma A, Chang J, et al. Wikiwhy: Answering and explaining cause-and-effect questions. ICLR 2023
>
> [3] Du L, Ding X, Xiong K, et al. e-CARE: a new dataset for exploring explainable causal reasoning. ACL 2022
>
> [4] Bai Y, Kadavath S, Kundu S, et al. Constitutional ai: Harmlessness from ai feedback. arXiv:2212.08073, 2022.
>
> [5] Yang K, Peng N, Tian Y, et al. Re3: Generating longer stories with recursive reprompting and revision. EMNLP 2022

---

### Meta-Review · Area_Chair_s6jb · 2023-09-19

**Recommendation:** 4

**Metareview:**

This paper introduces a collaborative framework for causality explanation generation where LLMs assume five roles: input processing, knowledge retrieval, and iterative answer refinement. The primary strength of the paper is the novelty of the framework, including bi-directional reasoning mechanism in the cause analyst and effect analyst. Indeed, such a collaborative framework can hold applications beyond causality in the future.

Several concerns related to empirical evaluation were addressed during the rebuttal phase. Please include the new empirical results in the camera ready. Additional ablations would further strengthen the paper, e.g., if cause and effect analysts do not communicate, what will be the impact.

In summary, this is an interesting and novel framework, with good experiments and analysis (incl. those conducted during the rebuttal).

---

### Decision · Program_Chairs · 2023-10-07

**Decision:**

Accept-Findings

**Comment:**

This paper introduces a collaborative framework for causality explanation generation where LLMs assume five roles: input processing, knowledge retrieval, and iterative answer refinement. The primary strength of the paper is the novelty of the framework, including bi-directional reasoning mechanism in the cause analyst and effect analyst. Indeed, such a collaborative framework can hold applications beyond causality in the future.

Several concerns related to empirical evaluation were addressed during the rebuttal phase. Please include the new empirical results in the camera ready. Additional ablations would further strengthen the paper, e.g., if cause and effect analysts do not communicate, what will be the impact.

In summary, this is an interesting and novel framework, with good experiments and analysis (incl. those conducted during the rebuttal).